# Analysis of Pneumonia Occurrence in Relation to Climate Change in Tanga, Tanzania

**DOI:** 10.3390/ijerph18094731

**Published:** 2021-04-29

**Authors:** Samweli Faraja Miyayo, Patrick Opiyo Owili, Miriam Adoyo Muga, Tang-Huang Lin

**Affiliations:** 1Department of Health and Nutrition, Tanga City Government, Tanga City 211, Tanzania; faraja088@gmail.com; 2Department of Public Health, School of Health Sciences, University of Eastern Africa, Baraton, Eldoret 30100, Kenya; owilip@ueab.ac.ke; 3Department of Human Nutrition and Dietetics, School of Medicine and Health Sciences, Kabarak University, Nakuru 20100, Kenya; mmuga@kabarak.ac.ke; 4Center for Space and Remote Sensing Research, National Central University, Taoyuan City 320, Taiwan

**Keywords:** pneumonia, climate change, rainfall, humidity, temperature, Tanga

## Abstract

In 2018, 70% of global fatalities due to pneumonia occurred in about fifteen countries, with Tanzania being among the top eight countries contributing to these deaths. Environmental and individual factors contributing to these deaths may be multifaceted, but they have not yet been explored in Tanzania. Therefore, in this study, we explore the association between climate change and the occurrence of pneumonia in the Tanga Region, Tanzania. A time series study design was employed using meteorological and health data of the Tanga Region collected from January 2016 to December 2018 from the Tanzania Meteorological Authority and Health Management Information System, respectively. The generalized negative binomial regression technique was used to explore the associations between climate indicators (i.e., precipitation, humidity, and temperature) and the occurrence of pneumonia. There were trend differences in climate indicators and the occurrence of pneumonia between the Tanga and Handeni districts. We found a positive association between humidity and increased rates of non-severe pneumonia (incidence rate ratio (IRR) = 1.01; 95% CI: 1.01–1.02; *p* ≤ 0.05) and severe pneumonia (IRR = 1.02; 95% CI: 1.01–1.03; *p* ≤ 0.05). There was also a significant association between cold temperatures and the rate of severe pneumonia in Tanga (IRR = 1.21; 95% CI: 1.11–1.33; *p* ≤ 0.001). Other factors that were associated with pneumonia included age and district of residence. We found a positive relationship between humidity, temperature, and incidence of pneumonia in the Tanga Region. Policies focusing on prevention and control, as well as promotion strategies relating to climate change-related health effects should be developed and implemented.

## 1. Introduction

The World Health Organization (WHO) [1] has identified pneumonia as the leading cause of mortality among children under 5 years old worldwide. Even though the global incidence of pneumonia among children and clinical pneumonia episodes have been found to have decreased by 30% and 22%, respectively, pneumonia alone still accounted for 15% of all deaths in children under 5 years old (i.e., over 800,000 deaths) in 2017 [1]. However, this was most prevalent in sub-Saharan Africa and Southern Asia. Furthermore, it remained the most common reason for hospitalization in sub-Saharan Africa [2,3]. 

In Tanzania, there have not been many studies that have focused on pneumonia [4,5,6,7]. These studies explored the hospital-based factors and not some of the other important contributors, such as environmental factors. Nevertheless, in 2016, pneumonia was considered one of the major health problems in Tanzania, which accounted for 15% of child mortality. In 2018, 70% of global pneumonia deaths occurred in about fifteen countries, with Tanzania being among the top eight countries [8]. In that same year, there were 17,624 under-5 years old deaths in Tanzania as a result of pneumonia infections. However, it still unknown if environmental factors contributed to the occurrence of pneumonia cases and death in Tanzania. 

Studies have shown that pneumonia can be spread in a number of ways, including either through the blood shortly after a child is born, or through the air from droplets of sneezing or coughing or through air pollution [1,9,10]. Several risk factors have also been found to contribute to the occurrence of pneumonia, such as the environmental factors [11,12,13,14], being on a ventilator or hospitalization, chronic diseases (i.e., lung disease, chronic obstructive pulmonary disease, or asthma), a weak immune system, and other comorbidities [15,16,17,18]. Of great importance to this study is the effect environmental factors on pneumonia occurrence. Seasonal, climatic, and weather conditions, which are affected by precipitation, humidity, and temperature, have been found to be associated with pneumonia occurrence and hospitalization [19,20,21,22,23,24]. The seasonal changes have also been found to contribute to the occurrence of particular pathogens, as more pneumonia cases are experienced in the cold months than in other seasons [25]. Chen et al. [26] used the Pneumoslide IgM test in children to explore mycoplasma pneumoniae, parainfluenza viruses, and respiratory syncytial, and found the viruses to be associated with seasonality. Mycoplasma pneumoniae was found to be higher in autumn, whereas Legionella pneumophila was found to be prevalent between summer and autumn. The adenovirus and influenza B virus were found to be prevalent in the months of summer and winter, respectively. Apart from climate indicators, there are also other factors that influence the susceptibility and severity of an individual, such as age, gender, community, secondhand smoke, air pollution, and childhood immunization [10,26,27,28,29,30].

Nevertheless, most of these studies were not conducted in low- and middle-income countries (LMICs), possibly because of data limitations. Yet, these LMICs contribute the greatest burden of disease globally. Tanzania, for example, is among the top eight countries with high pneumonia deaths, although it is still unknown whether environmental factors played a role pneumonia occurrence and deaths in Tanzania. Therefore, this study aimed to determine the trends in the climate change indicators and in the occurrence of pneumonia in the Tanga Region, Tanzania, and to subsequently explore the association between climate change and the occurrence of pneumonia. This study is significant because it contributes to the understanding of the effect of climate change on pneumonia occurrence and deaths in Tanzania in order to develop preventive and control measures. It may also contribute towards the development of early warning signs and adaptation techniques that would minimize the risk of respiratory illnesses and infectious disease associated with climate change. Moreover, this study is significant for the planning of health care resource allocation and for management of the effects of climate change.

## 2. Materials and Methods

### 2.1. Study Design and Data Sources

A time series study design was employed using meteorological and health data of the Tanga Region that were collected from January 2016 to December 2018. The data were obtained from the Tanzania Meteorological Authority (TMA) and Health Management Information System (HMIS), respectively. The data extracted from the TMA database included monthly average rainfall, humidity, temperature, and the year, while the data from HMIS included the number of individuals diagnosed with non-severe or severe pneumonia cases each month, age of the cases, district, and the year.

#### 2.1.1. Study Site

The Tanga Region is located in the far northeast corner of Tanzania, bordered by the Indian Ocean to the east, Pwani and Morogoro Regions to the south, Manyara Region to the West, and Kilimanjaro Region and Kenya to the north. The climatic conditions of the Tanga Region, both inland and in coastal areas, are predominantly warm and wet. However, the Handeni district, in the western plateau of the Usambara Mountains, has a generally hot and dry climate. The average temperature is about 26–29 °C and 30–32 °C in the hot months (between December and March) during the night and day, respectively. Meanwhile, the average temperature in cool months (between May and October) is about 20–24 °C and 23–28 °C during the night and day, respectively. In Tanga, the atmospheric humidity is relatively high, with a minimum of 65% and a maximum of 100%. With regards to precipitation, the region experiences an annual average of 750 mm of rainfall, where the coastal region of Tanga has an annual average of about 1100–1400 mm of rainfall.

#### 2.1.2. Sampling

A sample size of two districts was calculated from a total of 11 districts in the Tanga Region using an online Raosoft sample size calculator with a margin error of 5%, confidence level of 95%, and response distribution of 0.1%—because of the restrictions in acquiring government data. Typically, some data would not be accessible. The two districts were then randomly selected from the 11 districts using a simple random sampling technique. The data of the two districts selected (Tanga and Handeni) were then obtained and used in this study.

### 2.2. Variables

#### 2.2.1. Outcome Variable

The outcome variable, the count data of the monthly non-severe and severe pneumonia cases, collected from January 2016 to December 2018, was obtained from the HMIS database. The total number of diagnosed pneumonia cases from all of the health facilities are reported monthly to the HMIS office in the Tanga Region.

#### 2.2.2. Exposure Variables 

The three climate indicators that were included in this study were precipitation, humidity, and temperature. The daily averages from January 2016 to December 2018 were obtained from the TMA in the Tanga Region, before converting them into monthly averages. The temperature data were measured in degrees Celsius (°C), the data on the average precipitation were collected using rain gauges (measured in mm), and the data on humidity were collected using digital hygrometers (measured in percentage (%)).

#### 2.2.3. Other Variables

The other variables used in this study included the age group of the cases who had pneumonia (i.e., <1 month, 1 month to 1 year, 1 to 5 years, 5 to 60 years, and ≥61 years), district (i.e., Tanga and Handeni), and year (i.e., 2016, 2017, and 2018). The age, district, and year data were retrieved from the HMIS data.

### 2.3. Statistical Analysis

The data were analyzed in three phases. In the first phase, the descriptive statistics were analyzed in the form of the annual averages of the number of pneumonia cases and the meteorological, and these were presented in the form of means and standard deviations in a table. A two-sample *t*-test with equal variances estimation was used to test the mean differences between the Tanga and Handeni districts. The consistency of different procedures under unequal variance may vary greatly, hence the need to perform an equal variances test. 

In the second phase of the analysis, the monthly trends of the climate indicators and the non-severe and severe pneumonia cases were analyzed by age group and are presented using line graphs. The multilevel mixed-effects regressions with unstructured covariance structure—levels 1 and 2 being month and year, respectively—were used to test for trend differences between the Tanga and Handeni districts. As the climate indicators were continuous in nature, the Multilevel Mixed-Effect Linear Model technique was used to assess the trend differences between the two districts. However, the Multilevel Mixed-Effect Poisson Model was employed to analyze the trend differences of the discrete pneumonia data between the districts. The Multilevel Mixed-Effects Models were employed in order to adjust for the cluster and time effect of the data. We used the mixed and mepoisson STATA commands for the linear and Poisson models, respectively.

In the final phase of the analysis, the generalized negative binomial regression technique was employed to explore the associations between the climate indicators and the occurrence of pneumonia in the Tanga Region. The nbreg STATA command with the dispersion of the mean and the irr option was employed for the analyses. Hence, the incidence rate ratios (IRRs) were generated. Even though the negative binomial has a loose restrictive assumption, we explored both the unadjusted and the adjusted models to identify the effects of the potential confounders. The study analyzed two models, with the first being a crude model and the second being the adjusted analysis, which included all of the climate indicators, age groups, districts, and years. Analyses were performed separately for non-severe and severe pneumonia. The negative binomial regression was used because it can accommodate the over- and under-dispersion. The adjusted negative binomial regression equation is represented, as below:(1)$$\begin{array}{cc} Prob\left(\lambda =E\left(Y|X\right)\right) & \\ & =\beta_{0}+\beta_{1}\left(Rainfall\right)+\beta_{2}\left(Humidity\right)\\ & +\beta_{3}\left(Temperature_{max}\right)+\beta_{4}\left(Temperature_{min}\right)+\beta_{5}\left(Age\right)\\ & +\beta_{6}\left(District\right)+\beta_{7}\left(Year\right) \end{array}$$where the intensity parameter, *λ*, represents the expected number of pneumonia occurrence (*Y*) in a given period of time given a particular variable (*X*), and the *β*-coefficient is the difference between the log of expected counts, that is, *β* = log($\mu_{x}$ + 1) − log($\mu_{x}$), where *μ* is the expected count after adjusting for the predictor variable *x*. The exponential expression of *β*-coefficient, the IRRs, were used.

The crude and adjusted IRRs and 95% confidence intervals (CIs) were then presented in the table. The STATA version 13.1 software was used to analyze the data [31].

### 2.4. Ethical Considerations

For considering the ethical and scientific soundness, approvals were sought and obtained from the Research Ethics Committee of the University of Eastern Africa, Baraton (Kenya) and the National Research Ethics Committee (Tanzania). The Regional Medical Department of Tanga and the Tanzania Meteorological Authority also approved the study in Tanga.

## 3. Results

Table 1 presents the annual mean differences of pneumonia cases and the climate indicators between the Tanga and Handeni districts. There were no statistical differences between Tanga and Handeni among newborns (i.e., less than 1 month old) who had nonsevere pneumonia (*p*-values of 0.281 and 0.168 for 2017 and 2018, respectively) and severe pneumonia in all years (*p*-values of 0.090, 0.536, and 0.948 for 2016, 2017, and 2018, respectively). In 2017, the mean differences in the occurrence of severe pneumonia in Tanga and Handeni among all age groups were not statistically significant, except for infants (i.e., those aged between 1 month and 1 year), which was significantly different at *p* = 0.006. The mean difference among the elderly (i.e., aged 61 years and above) who had severe pneumonia was only statistically significant in the year 2018 (*p* = 0.011). 

### 3.1. Trends in the Climate Indicators

All of the other annual mean differences in the other age groups were statistically significant. The results also showed that the mean number of individuals with pneumococcal infections tended to be higher in the the Tanga district than in the Handeni district, across all of the considered years for almost all of the age groups. The annual mean rainfall differences between Tanga and Handeni were also not statistically significant throughout the years. The differences between all of the other climate indicators were statistically significant for all of the years, except for humidity levels in 2017 and 2018.

Only the trends in monthly rainfall (Figure 1A) and monthly minimum temperature (Figure 1D) were statistically different between the Tanga and Handeni districts, with the minimum temperature being much lower in Handeni than in Tanga. The difference in the maximum temperature (Figure 1C) was also statistically significant (*p* < 0.001). All of the graphs of the two districts exhibited a somewhat similar pattern in the trends of each indicator (i.e., each indicator in the two districts showed a similar upward and downward trend almost in the same month).

### 3.2. Trends in the Occurrence of Pneumococcal Infection

The monthly trend in the number of non-severe and severe pneumonia infections among children under 5 years old is shown in Figure 2. The rates of non-severe infections were higher in the Tanga district than in the Handeni district, except for the non-severe infections among the newborns below the age of 1 month old, which had some months higher in the Handeni compred with Tanga district. This trend was also similar among children below 1 month old who had severe pneumonia. However, the rate of severe infection seemed to peak to the highest level in Tanga (Figure 2B,D) between the months of March and April 2017, and in Handeni in the month of June 2017 (Figure 2F). All of the trend differences between the Tanga and Handeni districts were statistically significant.

Figure 3 shows the trend differences in pneumonia infection among children aged 5 years old and above and in adults. The infection trend differences between the Tanga and Handeni districts were statistically significant in all of the age groups. Pneumonia infections seemed to be higher in Tanga across different months of different years in all of the age groups. However, only the Handeni district exhibited a much higher rate of severe infection among children and adults aged 5–60 years old (Figure 3B) compared with Tanga in June 2017.

### 3.3. Association between Climate Change and Pneumococcal Infections

The crude and adjusted negative binomial regression results of the relationship between the climate indicators and pneumonia infection are presented in Table 2. The only climate indicator that was positively associated with non-severe pneumonia after adjusting for all variables was humidity with an increased rate of infection (IRR = 1.01; 95% CI: 1.01–1.02; *p* ≤ 0.05). There was an increased rate of non-severe pneumonia among those who were younger than 60 years old. Those aged between 1–5 and 5–60 years old had 3.80 (95% CI: 3.35–4.30; *p* ≤ 0.001) and 3.69 (95% CI: 3.26–4.18; *p* ≤ 0.001) times higher rates of non-severe pneumonia, respectively, than those who were older than 60 years. Only infants (i.e., those below 1 month) had a lower incidence rate of both non-severe and severe pneumonia than the elderly (i.e., aged 61 and above), yet the rates of non-severe pneumonia in the Handeni district were 49% lower than that of Tanga district after adjusting for all of the variables.

However, after adjusting for all of the variables, an increase in the levels of humidity and the minimum temperature increased the rates of severe pneumonia at IRR = 1.02 (95% CI: 1.01–1.03; *p* ≤ 0.05) and IRR = 1.21 (95% CI: 1.11–1.33; *p* ≤ 0.001), respectively, while the increased maximum temperature reduced the rate of pneumonia infection by 14%. Similarly, those aged between 1–5 and 5–60 years old had higher rates of severe pneumonia than those aged above 60 years old at IRR = 5.37 (95% CI: 4.33–6.67; *p* ≤ 0.001) and IRR = 3.71 (95% CI: 2.99–4.60; *p* ≤ 0.001), respectively. Again, the Handeni district had a significantly lower incidence rate of severe pneumonia than the Tanga district. Nevertheless, the incidence rate of severe pneumonia in the year 2017 was 26% higher than that in 2016 (95% CI: 1.05–1.52; *p* ≤ 0.05).

## 4. Discussion

In this study, we assessed the occurrence and trend of non-severe and severe pneumonia among different age groups before exploring the associations between climate indicators and pneumonia in the Tanga Region, Tanzania. The analyses revealed that there was a mean difference in the occurrence and trend of pneumonia in the Tanga and Handeni districts. This supports the results of other studies on the regional and seasonal variability of pneumonia [19,32]. However, for the association between the climate indicators and rate of pneumococcal infection, we found a positive association between humidity and non-severe and severe pneumonia. Minimum temperature was also positively associated with severe pneumonia, whereas maximum temperature had a negative association. The findings of this study are in agreement with those of other authors, who found that humidity and temperature are contributing factors to the occurrence of pneumonia [19,20,21,22,33].

Nevertheless, the interaction between temperature and humidity remains contentious in the literature. Some studies have found that low temperatures with low humidity increase the risk of pneumonia [23,34]. Other studies have found a different result—that high temperatures and high humidity increase the rate of pneumonia [22,24]. In contrast to these studies, we found that low temperature and high humidity levels increased the rates of pneumonia. Some of the possible reasons for this finding could be as a result of other potential confounders that may be associated with the individual cases that have not been explained in this study, such as biological factors. The temperature–humidity interaction phenomenon may weaken the nasal defense mechanisms and/or increase the survival of respiratory viruses [34], depending on other factors that were measured here. Our study, however, found a positive association between cold temperatures, high humidity, and the occurrence of pneumonia. Several studies conducted elsewhere have also found that the incidence of respiratory infection increased with a decrease in air temperature [11,35]. Conversely, a study in two subtropical Chinese cities found that exposure to high temperatures could cause acute and chronic health effects [35]. The contradictions in the interaction between temperature and humidity should still be explored still in order to understand the microorganism activity during different weather conditions. 

The nonsignificant association found between rainfall and pneumonia in this study was also contrary to the findings of other studies, which have alluded to the occurrence of pneumonia during rainy seasons [12,13]. The possible explanation of this finding could be as a result of the spatial variations in different geographical locations, although this study supports the findings of other authors who found a nonconvincing relationship between rainfall and pneumonia [14,36]. Only temperature and humidity changes had an effect on pneumonia, just as was reported by others [10], while precipitation may possibly have an influence on the viral activity and the transmission of the virus [12]. 

Even though this study documented an interesting difference in the occurrence of pneumococcal infection across different age groups—with nonstatistical differences between the Tanga and Handeni districts among newborns (aged less than 1 month old) in 2017 and 2018—the adjusted IRR revealed that newborns (aged less than 1 month) were 77% and 57% less likely to acquire non-severe and severe pneumonia, respectively, than the elderly (aged over 60 years). All other age groups (i.e., between 1 month and 60 years old) were more likely to acquire both non-severe and severe pneumonia than the elderly. Age was found to have a strong influence on the general incidence of invasive infections. However, our findings do not support another study, which found that invasive pneumonia is most frequent in the first years of life and in persons older than 65 years, who remain vulnerable to pneumonia-related morbidity and mortality [28]. These might possibly be as a result of several other factors, such as the weak immune systems of infants and the elderly or from exposure to secondhand tobacco smoke or air pollution [10,29]. Nevertheless, the findings herein indicate that pneumonia was more common among children between the age of between 1 month and 5 years than in the elderly aged 60 years and above. We expected a decrease in invasive pneumococcal disease among infants (less than 1 year) and children under five, because of the introduction of a pneumococcal conjugate vaccine at 6, 10, and 14 weeks, yet they were more likely to acquire pneumonia than the elderly. 

The Centers for Disease Control and Prevention (CDC) recommends a pneumococcal vaccination for all children younger than 2 years old, those older than 2 years with certain medical conditions, and the elderly (aged over 65 years). Therefore, it could be deduced that the higher rates of pneumonia infection among those aged between 1 month and 60 years old could be the result of individuals who either have certain medical conditions, such as chronic lung diseases (e.g., chronic obstructive pulmonary disease, bronchiectasis, or cystic fibrosis), which make the lungs more vulnerable, or those who are not vaccinated against pneumonia. Nonetheless, the effect of the pneumococcal vaccine still needs to be explored further, as infection can either be transmitted through vertical or horizontal means. Moreover, the individuals in the data may have exhibited other medical conditions that make the lungs vulnerable, or were predisposed to other factors that could lead to pneumonia [9,29,30,37,38].

Our study also found that the differences in the pneumococcal infection rate between the two districts were statistically significant, with the Handeni district exhibiting a much lower rate of infection than the Tanga district. This is likely a result of either the differences in the topographical nature or the population density of the two districts, with the population in the Tanga district presumably being much higher than that in the Handeni district. The occurrence of severe pneumonia was also higher in 2017 than 2016, after adjustment. However, pneumonia cases in 2018 were lower than those in 2016; however, this was not statistically significant. The reduction in 2018 may have been due to other reasons, such as favorable weather conditions, knowledge of the prevention techniques, use of home remedies, or antibiotic usage, although these need to be explored further. Therefore, strategies to improve the case management of pneumonia are needed in order to improve the prevention and control of pneumonia and, subsequently, the reduction of pneumonia-specific morbidity and mortality. 

In this study, a positive relationship between humidity and non-severe or severe pneumonia was found, where the relationship between the minimum temperature and severe pneumonia was also positive. However, no association between precipitation and pneumonia was observed.

### Limitations and Strengths

One main limitation of this study was the assumption that all pneumococcal infections were a result of climate indicators. These infections might have been the result of other known predisposing factors. Secondly, this study had data limitations; therefore, we were unable control for the potential individual factors that might be associated with pneumonia. Thirdly, we were unable to assess the interaction effect and the dose–response relationship between the climate indicators and mortality. Finally, the point estimate may not have reflected the true causality, because of the ecological nature of the study (i.e., ecological data from 2016 to 2018); hence, the data should be interpreted with caution. There could have been an underestimation or overestimation of the true effect. Therefore, in order to extend the findings of this study, future studies should adequately explore the actual exposures and predisposing factors, following-up and exploring the interactions. Nonetheless, we attempted to explore the relationship between climate indicators and pneumococcal infection in Tanzania, becoming the first study to do so. Secondly, obtaining the actual climate indicators and the actual reported pneumonia cases from the Tanzania Meteorological Authority (TMA) and Health Management Information System (HMIS), respectively, was an important strength of this study. There is generally inadequacy in data collection in low- and middle-income countries and, in most cases, relevant data are not available. Finally, the analytical technique used is also a strength, as it provided a clear method for assessing the relationship between climate indicators and the incidence of pneumonia.

## 5. Conclusions

In conclusion, we explored the association between climate change and the occurrence of pneumonia, and we found a positive relationship between humidity and temperature and the occurrence of pneumonia in the Tanga Region, Tanzania. Even though this ecological study provided a temporal causality, it can serve as a point of reference for other studies that might be conducted in Tanzania, in terms of the relationships between climate and health outcomes. Prevention and control strategies should be developed, in order to mitigate the effects of climate change and the associated health effects. Health promotion for and population sensitization to prevention and control strategies are also necessary in order to reduce the incidence of pneumonia in the Tanga Region. The development of early warning systems and adaptation strategies can possibly help minimize the risk of climate-sensitive infectious diseases such as pneumonia. Infectious disease surveillance should also be improved by the health department in order to identify the disease pattern and to plan for equitable distribution of resource in different seasons and between different regions. The study has contributed to a better understanding of the association between climate change and the occurrence of pneumonia, especially in Tanzania. Climate change is an issue of concern in many countries globally, Tanzania included. Thus, this study has contributed into the body of knowledge on climate change and pneumonia, especially in LMICs. It has also formed a useful foundation on which other researchers can undertake further studies on the health effects of climate change in Tanzania, and most importantly, Africa.

## Figures and Tables

**Figure 1 ijerph-18-04731-f001:**
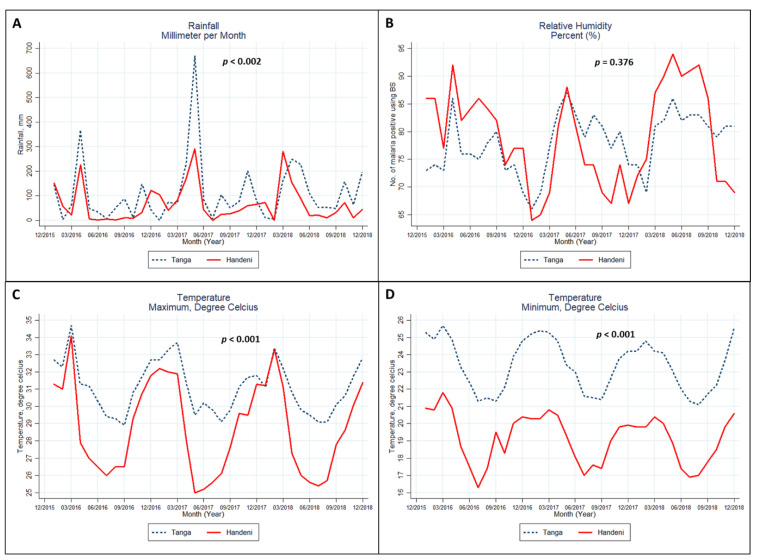
Monthly rainfall (**A**), humidity (**B**), maximum temperature (**C**), and minimum temperature (**D**) from 2016 to 2018 in Tanga, Tanzania. Note: multilevel linear mixed-effects regression—with levels 1 and 2 being month and year, respectively—was used to test the trend differences.

**Figure 2 ijerph-18-04731-f002:**
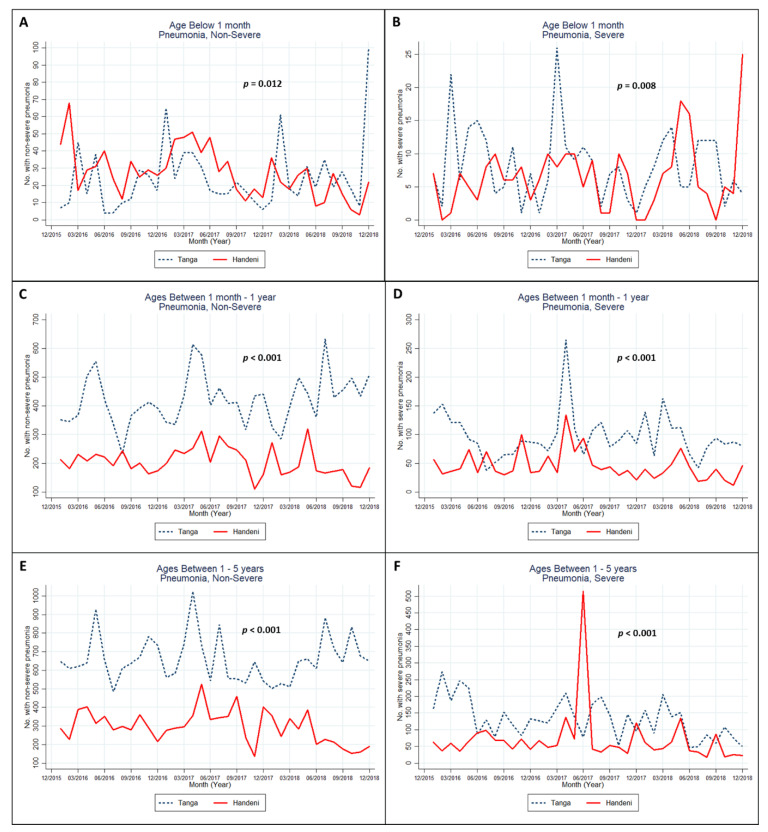
Monthly non-severe (**A**,**C**,**E**) and severe (**B**,**D**,**F**) pneumococcal infections in children under five from 2016 to 2018 in Tanga, Tanzania. Note: multilevel mixed-effects Poisson regression—with levels 1 and 2 being month and year, respectively—was used to test the trend differences.

**Figure 3 ijerph-18-04731-f003:**
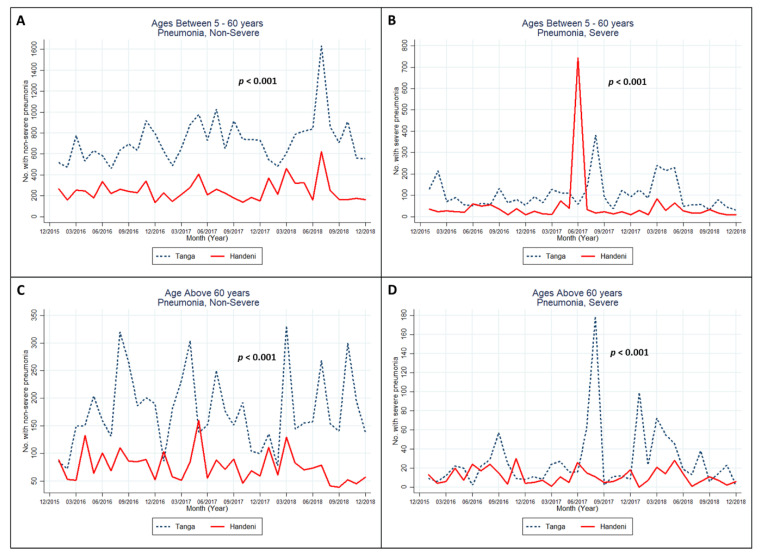
Monthly non-severe (**A**,**C**) and severe (**B**,**D**) pneumococcal infections of the age groups of 5 years old across 2016 to 2018 in Tanga, Tanzania. Note: multilevel mixed-effects Poisson regression—with levels 1 and 2 being month and year, respectively—was used to test the trend differences.

**Table 1 ijerph-18-04731-t001:** The annual mean differences of pneumonia infections and climate variables between the Tanga and Handeni districts by year.

	2016, Annual *µ* (*SD*) ^a^			2017, Annual *µ* (*SD*) ^a^			2018, Annual *µ* (*SD*) ^a^		
	Tanga	Handeni	*p*-Value	Tanga	Handeni	*p*-Value	Tanga	Handeni	*p*-Value
Nonsevere Pneumonia									
<1 month	18.1 (13.5)	31.6 (14.5)	0.027	25.1 (16.3)	32.1 (14.7)	0.281	30.1 (25.9)	18.6 (10.4)	0.168
1 month–1 year	389.8 (81.6)	203.9 (25.8)	<0.001	431.8 (89.7)	227.7 (55.6)	<0.001	438.8 (92.5)	185.0 (57.1)	<0.001
1–5 years	668.4 (108.8)	308.2 (59.1)	<0.001	655.3 (154.1)	334.1 (101.3)	<0.001	655.0 (116.9)	244.5 (79.5)	<0.001
5–60 years	638.2 (139.2)	239.9 (62.1)	<0.001	763.8(158.0)	219.7 (73.9)	<0.001	775.8 (306.7)	283.3 (145.7)	<0.001
≥61 years	176.2 (69.4)	81.7 (25.1)	<0.001	172.1 (65.2)	77.8 (31.3)	<0.001	182.8 (76.3)	70.0 (27.7)	<0.001
Severe Pneumonia									
<1 month	8.8 (6.1)	5.3 (3.0)	0.090	7.8 (6.8)	6.4 (3.8)	0.536	8.1 (4.1)	7.9 (7.7)	0.948
1 month–1 year	92.2 (34.9)	48.4 (22.1)	0.001	107.5 (52.3)	54.1 (32.2)	0.006	93.3 (33.6)	35.3 (17.7)	<0.001
1–5 years	156.3 (65.5)	61.9 (20.0)	<0.001	137.8 (46.9)	101.8 (134.2)	0.390	101.8 (51.6)	49.0 (34.0)	0.007
5–60 year	88.8 (48.4)	32.8 (16.6)	0.001	119.0 (87.8)	85.9 (208.0)	0.617	104.3 (79.7)	29.0 (23.1)	0.005
Climate Indicators									
Rainfall, _mm_	83.6 (101.2)	53.1 (73.5)	0.406	136.6 (180.7)	78.8 (80.1)	0.322	111.2 (85.1)	66.9 (80.4)	0.204
Humidity, _%_	75.6 (4.3)	82.3 (5.2)	0.002	78.3 (6.2)	72.8 (7.4)	0.058	80.2 (4.5)	82.3 (9.8)	0.493
Temperature, _Max °C_	31.3 (1.7)	29.1 (2.7)	0.024	31.2 (1.5)	28.7 (2.8)	0.012	30.9 (1.4)	28.6 (2.7)	0.020
Temperature, _Min °C_	23.5 (1.7)	19.4 (1.7)	<0.001	23.5 (1.5)	19.2 (1.3)	<0.001	23.2 (1.5)	18.9 (1.3)	<0.001

^a^ Mean differences between districts were analyzed using the two-sample *t*-test technique with equal variances assumption.

**Table 2 ijerph-18-04731-t002:** Annual mean differences of pneumonia infections and climate variables between the Tanga and Handeni districts by year.

Infectious Disease	Crude IRR (95% CI)	Adjusted IRR (95% CI)
Nonsevere Pneumonia		
Rainfall, _mm_	1.01 (1.01, 1.02) ***	1.00 (1.00, 1.00)
Humidity, _%_	1.01 (0.99, 1.02)	1.01 (1.01, 1.02) **
Temperature, _Max °C_	1.07 (1.02, 1.12) ***	1.00 (0.96, 1.04)
Temperature, _Min °C_	1.13 (1.10, 1.18) ****	1.01 (0.96, 1.06)
Age group (ref: >60 years)		
<1 month	0.20 (0.17, 0.24) ****	0.23 (0.20, 0.27) ****
1 month–1 year	2.47 (2.08, 2.93) ****	2.52 (2.22, 2.85) ****
1–5 years	3.77 (3.17, 4.47) ****	3.80 (3.35, 4.30) ****
5–60 years	3.84 (3.23, 4.56) ****	3.69 (3.26, 4.18) ****
District (ref: Tanga)		
Handeni	0.42 (0.35, 0.51) ****	0.51 (0.43, 0.60) ****
Year (ref: 2016)		
2017	1.07 (0.83, 1.37)	1.07 (0.96, 1.20)
2018	1.05 (0.81, 1.34)	0.95 (0.86, 1.05)
Severe Pneumonia		
Rainfall, _mm_	1.01 (1.01, 1.02) **	1.00 (1.00, 1.00)
Humidity, _%_	1.02 (0.99, 1.03) *	1.02 (1.01, 1.03) **
Temperature, _Max °C_	1.02 (0.97, 1.07)	0.86 (0.80, 0.92) ****
Temperature, _Min °C_	1.11 (1.07, 1.16) ****	1.21 (1.11, 1.33) ****
Age group (ref: >60 years)		
<1 month	0.38 (0.29, 0.49) ****	0.43 (0.34, 0.54) ****
1 month–1 year	3.67 (2.84, 4.73) ****	3.80 (3.06, 4.72) ****
1–5 years	5.18 (4.02, 6.68) ****	5.37 (4.33, 6.67) ****
5–60 years	3.91 (3.03, 5.05) ****	3.71 (2.99, 4.60) ****
District (ref: Tanga)		
Handeni	0.50 (0.40, 0.62) ****	0.73 (0.54, 0.98) **
Year (ref: 2016)		
2017	1.26 (0.95, 1.66)	1.26 (1.05, 1.52) **
2018	0.90 (0.68, 1.18)	0.94 (0.79, 1.11)

IRR—incidence rate ratio; CI—confidence interval. * *p* ≤ 0.10; ** *p* ≤ 0.05; *** *p* ≤ 0.01; **** *p* ≤ 0.001.

## Data Availability

All relevant data are within the paper or can be requested from the Tanzania Meteorological Authority (TMA) and the Health Management Information System (HMIS) of the Republic of Tanzania. These are the competent authorities to approve data accessibility.

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
