# Peer review of "Analysis of Pneumonia Occurrence in Relation to Climate Change in Tanga, Tanzania"

_ijerph, 2021, doi:10.3390/ijerph18094731_

Round 1

Reviewer 1 Report

  1. The introduction perspective is not wide enough. I suggest that authors should include more statistics of pneumonia.
  2. The authors have not discriminated between the relevant and irrelevant material.
  3. Explain or show why your study is a longitudinal study. Did your data involve repeated observations of the same variables? If so, include the variables that their observations collected repeatedly.
  4. Your sample size calculation is very wrong and unprofessional. Sample size is calculated from the population size and 11 districts is not a population size. How did you get 2 districts from 11?
  5. Your outcome variable is a count data and count data are continuous data. So, your study is not longitudinal in nature.
  6. There is no need to separate the variables into outcome, exposure, and other variable. Describe all of them under same name.
  7. In line 159, what do you mean by marginally significant at p = 0.072? its either significant or not.
  8. In line 160, I cannot see a somewhat similar pattern in your trends. You need to interpret your trend accordingly such as if its upward, downward, cyclical, or irregular and explain which of the months/year are the values considered to influence the pneumonia cases.
  9. Though, you mentioned multilevel mixed-effects Poisson regression with unstructured covariance structure but was not used or applied. However, Negative binomial regression is a generalization of Poisson regression which loosens the restrictive assumption that the variance is equal to the mean made by the Poisson model. So, I suggest authors should mention why they use negative binomial regression.
  10. I want to see the mathematical expression used for the negative binomial regression and a brief step taken in analyzing your data.
  11. The results summarized in Table 2 is not the results for negative binomial regression but Incidence rate ratio (IRR), which is the ratio of two incidence rates. The ratio between two cumulative incidences (risk in exposed divided by risk in unexposed) gives the relative risk (or risk ratio).
  12. Your analysis is not sufficiently analytical. I do not trust it.
  13. In your discussion (line233) you mentioned that high temperature had a negative association. Where is this result presented in your plot/table?
  14. Authors did not discuss anything on other part of analysis, only on the trend analysis.

Reviewer 2 Report

Please find attached files

Round 2

Reviewer 1 Report

The revised manuscript completely satisfies the request of my original comments. It seems to me that the methods are fully sounding, and the conclusions are in line with what are currently considered the most effective solutions for preventing recurring waves of Pneumonia in Tanga, Tanzania. However, I suggest authors attend to the following:

  1. My comment 2: I was referring to your introduction and authors have justified it by adding more relevant materials. The introduction section completely satisfies the comment one and two.
  2. Comment 3: I am not satisfied with the authors’ reply. I think your study design is not longitudinal but time series data. However, I suggest authors should check the difference between time series and longitudinal data. Let me give you an instance, pneumonia of an individual over a period of ten years is an example of time series data while pneumonia of set of individuals over a period of ten years is an example for longitudinal data. Cross check your data for a proper study design. Meanwhile, I would love to see your data for cross-checking.
  3. Comment 4: Authors’ reply is baseless and not statistically analytical. What sample size calculation does is extracting a subset (sample) from a population be it individual data or metrological data. Your sample size calculation is wrong. I used the Raosoft and still did not get 2 districts with the information provided in the manuscript. I suggest authors remove the sample size calculation from their design and explain why they chose the two districts used in the manuscript.
  4. Comment 5: I still do not agree with the authors. Count data are discrete data, yes but for the variables in your results, they are not count data. Authors mentioned in page 3, line 125 that “the outcome variable, the count data of the month….” is collected for the study and I was saying that your data variable should not be count data, discrete data, or longitudinal data but continuous data. For example: Continuous Data information can be measured on a continuum or scale compare to discrete data that can be measured like good or bad, off or on, etc., continuous data can be recorded at many different points such as length, size, width, time, temperature, humidity, precipitation etc., which is what were used in this manuscript. This is not an attack on anyone personality(ies) because I have no conflict of interest.

Author Response

Please see the attachment, thanks.

Reviewer 2 Report

The authors have dealt with the most of comments raised by the reviewers. The paper also needs the work of a copy editor or a proofreader.

Author Response

Thank you. The manuscript has been copy edited for typos and grammatical errors.